

# AMCFCN: attentive multi-view contrastive fusion clustering net

Huarun Xiao, Zhiyong Hong, Liping Xiong and Zhiqiang Zeng

College of Electronic and Information Engineering, Wuyi University, Jiangmen, Guangdong, China

## ABSTRACT

Advances in deep learning have propelled the evolution of multi-view clustering techniques, which strive to obtain a view-common representation from multi-view datasets. However, the contemporary multi-view clustering community confronts two prominent challenges. One is that view-specific representations lack guarantees to reduce noise introduction, and another is that the fusion process compromises view-specific representations, resulting in the inability to capture efficient information from multi-view data. This may negatively affect the accuracy of the clustering results. In this article, we introduce a novel technique named the "contrastive attentive strategy" to address the above problems. Our approach effectively extracts robust view-specific representations from multi-view data with reduced noise while preserving view completeness. This results in the extraction of consistent representations from multi-view data while preserving the features of view-specific representations. We integrate view-specific encoders, a hybrid attentive module, a fusion module, and deep clustering into a unified framework called AMCFCN. Experimental results on four multi-view datasets demonstrate that our method, AMCFCN, outperforms seven competitive multi-view clustering methods. Our source code is available at https://github.com/xiaohuarun/AMCFCN.

## INTRODUCTION

In real-world scenarios, various heterogeneous visual features are commonly encountered, including HOG (*Dalal & Triggs, 2005*), SIF (*Lowe, 2004*), and LBP (*Ojala, Pietikäinen & Harwood, 1996*). Multi-view data is data that can be described in many different ways for the same thing, but which all share the same clustering structure. As a result, multi-view clustering (*Li et al., 2019*; *Yang et al., 2021*; *Lin et al., 2023*) garners significant attention in the machine-learning community. In the past decades, several traditional clustering methods were proposed, such as non-negative matrix factorization (*Chen et al., 2020a*; *Chen et al., 2020b*), subspace clustering (*Cao et al., 2015*), and spectral clustering (*Li et al., 2018*). However, despite the effectiveness of traditional clustering methods for single-view data, their applicability to multi-view data is limited due to the underutilization of a shared value representation of multi-view (or as view-common representation). Thus, the multi-view fusion technique becomes crucial in the context of multi-view clustering (*Ke et al., 2021*; *Ke, Zhu & Yu, 2022*).

Corresponding author
Zhiyong Hong, hongmr@163.com

In previous studies, exemplified by reconstruction fusion-based models (*Zhang et al., 2020*; *Ke et al., 2022*), the primary focus lies on the extraction of multiple view-specific representations and the subsequent acquisition of a shared view-common representation, which serves the purpose of reconstructing the original data. The reconstruction process mandates the network to capture details, potentially introducing noise into the learned representation (*Ke, Zhu & Yu, 2022*). Consequently, the network's ability to effectively extract both view-common and view-specific representations during the fusion process may be compromised, while there are some approaches (*Li et al., 2019*; *Zhou & Shen, 2020*) that use GAN (*Mirza & Osindero, 2014*) to improve the stability of view-specific representations, as well as constrain the model to avoid compromising view-specific representations during the fusion process (over-fusion). We observe that this can result in prolonged training times and an unstable training process.

In this article, we identify two challenges for multi-view clustering: (i) how to obtain a representation with less noise in a large amount of unlabeled data. (ii) when fusing multiple views to obtain view-common representation for clustering, it is important to preserve view-specific completeness. For the first challenge, we employ the attention mechanism as a solution. The attention mechanism plays a crucial role in numerous state-of-the-art neural architectures (*Guo et al., 2022*), enabling models to dynamically concentrate on specific parts of inputs for the identification of the most informative and discriminative features from each view (*Lu, Liu & Zuo, 2021*). For the second challenge, instead of allowing view-specific representations to interact directly with the view-common representation, we introduce an additional alignment space for view-specific alignment interactions, hereby safeguarding the integrity of the representation.

In brief, we argue that good view-common representations require less noise and completeness. Inspired by the substantial achievements of contrastive learning (*Chen et al., 2020b*) in the context of multi-view learning (*Ke et al., 2021*; *Ke et al., 2023*), we propose a novel framework of Attentive Multi-View Contrastive Fusion Clustering Net (AMCFCN). This framework employs view-specific encoders and a fusion module to extract robust representation from multi-view data. Additionally, we introduce a clustering-guided mechanism to drive the fusion module to learn the view-common representation. Unlike previous approaches (*Lin et al., 2021*; *Ke et al., 2022*), we introduce a novel contrastive strategy, called contrastive attentive strategy (COATS), for maintaining view-specific representations. The main contributions of this article are summarized as follows:

(1) We propose a hybrid attentive module that combines the dynamic contribution of view-specific representations to representation learning to derive robust representations.

(2) We introduce a contrastive attentive strategy (COATS) for view fusion while preserving view-specific completeness.

(3) We integrate view-specific encoders, a hybrid attentive module, a fusion module, and deep clustering into a unified framework known as AMCFCN. We conduct experiments on four clustering datasets to measure the clustering performance of AMCFCN. A large number of experiments validate the effectiveness of the proposed method.

## RELATED WORK

In this section, we briefly review multi-view clustering. Existing multi-view clustering methods fall into three categories: traditional multi-view clustering methods (*Cao et al., 2015*; *Liu et al., 2016*; *Chen et al., 2020a*), deep learning-based methods (*Li et al., 2019*; *Zhou & Shen, 2020*), and deep learning methods combining with traditional multi-view clustering techniques (*Gao et al., 2020*; *Yang et al., 2021*).

Traditional multi-view clustering methods can be classified into four categories, including those based on kernel learning (*Liu & Tuzel, 2016*), subspace learning (*Cao et al., 2015*), non-negative matrix decomposition (*Chen et al., 2020a*), and spectral approaches (*Li et al., 2018*). For example, the subspace learning method (*Cao et al., 2015*) focuses on dimensionality reduction across multiple views and information fusion, aiming to reveal intricate data structures and improve clustering outcomes. The non-negative matrix decomposition method (*Chen et al., 2020a*) aims to identify common basis vectors for each view, with their similarities influencing the clustering effectiveness. However traditional methods struggle with high-dimensional and massive data in the big data era.

Now, deep learning is gaining popularity due to its powerful nonlinear and parallel computing capabilities. Deep learning-based approaches harness deep neural networks to learn discriminative features from each view and extract view-common representation for improved clustering. In the following, we present four of the latest and best deep learning-based multi-view clustering, which will be used to compare with our method. DAMC (*Li et al., 2019*) deploys deep autoencoders to facilitate the efficient transformation of raw features into a shared, low-dimensional embedding space, simultaneously acquiring shared representation across multiple views. It also employs adversarial training to capture data distributions and decompose mappings. EAMC (*Zhou & Shen, 2020*) employs adversarial learning and attention mechanisms to redistribute feature distributions and to elucidate the significance of three tasks: feature learning, modal fusion, and clustering assignment. Ae2-Nets (*Zhang, Liu & Fu, 2019*) combines view-specific representations learning with multi-view information encoding using a nested auto-encoder framework, achieving a balance between complementarity and consistency among multi-views. CONAN (*Ke et al., 2021*) extracts consistent representations from multiple views, preserves view-specific features, and achieves view alignment through contrastive learning to obtain a good view-common representation.

In contrast, there exist some deep learning methods combined with traditional multi-view clustering techniques (*Gao et al., 2020*; *Yang et al., 2021*). Deep Canonical Correlation Analysis (DCCA) (*Gao et al., 2020*), as an alternative to kernel canonical correlation analysis (*Liu & Tuzel, 2016*), is recognized for its scalability with data size. It is introduced as a method for acquiring correlated representations by learning nonlinear transformations from two data views. Deep multimodal subspace clustering (DMSC) (*Yang et al., 2021*) utilizes subspace learning (*Cao et al., 2015*) and measures the loss during training by assessing the distance between the reconstruction and the original input, achieving a view-common representation through spatial fusion. Simultaneously, there exist deep learning methods combined with traditional k-means clustering methods (*Caron et al.,*

*2018*). For example, *Caron et al. (2018)* propose a sequential paradigm that separates the two key tasks of feature learning and clustering, and can only perform clustering on the entire dataset.

However, the above methods also have room for improvement. For instance, in the context of deep learning, DAMC (*Li et al., 2019*) and CONAN (*Ke et al., 2021*) neglect the noise in view-specific representations, potentially yielding suboptimal results. EAMC (*Zhou & Shen, 2020*) employs an attention mechanism to reduce representation noise but uses GAN for model constraints, which may lead to model instability. On the other hand, *Caron et al. (2018)*, the authors combine deep learning and K-Means while separating the two crucial tasks of feature learning and clustering. This separation has the potential to delay the network's training in the optimal direction. Our AMCFCN effectively reduces representation noise with the attention mechanism while preserving view integrity through COATS. Additionally, it seamlessly integrates feature learning and clustering, co-optimizing both tasks. This adaptation allows the clustering process to improve as the feature learning process unfolds, ultimately enhancing clustering results.

## METHOD

In this section, firstly, we introduce the formulation of the problem. This is followed by an introduction to the hybrid attentive module and the contrastive attentive strategy. Lastly, we present the objective function.

### Formulation of the problem

In this section, we introduce the goal of our study. The objective of AMCFCN is to go from a dataset consisting of N data samples, which comprises Q views $\{M^{(1)}, M^{(2)} \cdots, M^{(Q)}\}$, where $M^{(i)} \in \mathbb{R}^{N \times d(M^{(i)})}$ represents data samples of the dimension $d(M^{(i)})$ from the i-th views. The aim is to derive a set of view-common representations denoted as G and subsequently cluster the N data samples into c clusters using this view-common representation. The proposed method AMCFCN consists of Q view-specific encoder networks $E(\cdot)$, Q hybrid attentive module $A(\cdot)$, a fusion module, and a clustering head $C(\cdot)$ as illustrated in Fig. 1.

The AMCFCN contains Q view-specific encoder networks $E(\cdot)$, which extracts view-specific feature maps $\{X^{(i)}{}_{i=1}^{Q}\}$ from multi-view data $\{M^{(i)}{}_{i=1}^{Q}\}$; $A_i(\cdot)$ is utilized to obtain view-specific robust representations $\{Y^{(i)}{}_{i=1}^{Q}\}$ from these view-specific feature maps $\{X^{(i)}{}_{i=1}^{Q}\}$; $f(\cdot)$ is responsible for generating the view-common representation by amalgamating the view-specific representations into G. In COATS, we optimize the view-common representation G by taking G and $\{Y^{(i)}{}_{i=1}^{Q}\}$ as inputs. Finally, $C(\cdot)$ applies off-the-shelf clustering methods on G to evaluate the effectiveness of the view-common representation G. Below we describe the functions of each module in detail.

### *View-specific encode network*

$E(\cdot)$: The survey (*Zhu et al., 2023*) provides a summary of feature extraction methods applicable to multimodal domains, encompassing text, vision, and audio. The survey (*Zhu et al., 2023*) recommends the utilization of a Convolutional Neural Network (CNN) for feature extraction. CNN is preferred due to its capacity to alleviate the laborious process

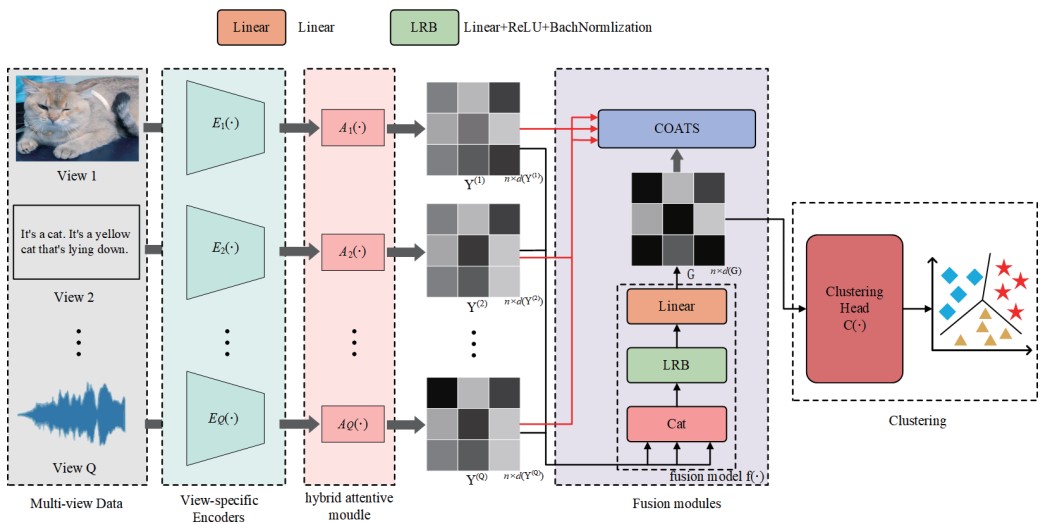

**Figure 1  Illustration of the workflow of AMCFCN.**

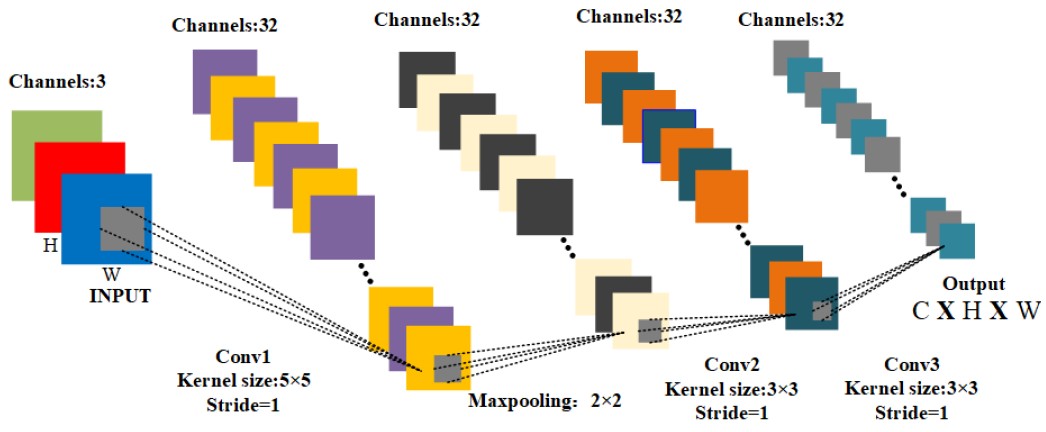

**Figure 2  The network structure of the view encoder.**

of manual feature extraction and its effectiveness in extracting features from images. The network structure diagram of CNN is depicted in Fig. 2. $E(\cdot)$ transforms the input image data $M^{(i)Q}_{i=1}$, which is converted into the feature map type $\{X^{(i)Q}_{i=1}\} \in \mathbb{R}^{C \times H \times W}$, as follows:

$$X^{(i)} = E^{(i)}\left(M^{i}; W_E^{(i)}\right), \text{where } i = 1, 2, \ldots, Q \tag{1}$$

here, $W_E^{(i)}$ denotes the weight of the i-th view encoder $E^{(i)}(\cdot)$.

### Hybrid attentive module

$A(\cdot)$**:** $A(\cdot)$ acts to allow the network to weight the view-specific feature map X into HA, *i.e.,* to allow the network to adapt itself to the part of its interest (also known as the downstream task-relevant information). Then, the obtained feature map information is subjected to dimensionality reduction. We perform the dimensionality reduction using the

dimensionality reduction module T(), which maps all view-specific feature maps that have been undergone by the hybrid attentive module to a lower dimensional space, *i.e.,* maps robust view-specific representations Y. The next section goes into more detail, $HA^{(i)}$ and $Y^{(i)}$ are defined as follows:

$$HA^{(i)} = A\left(X^{(i)}\right) \tag{2}$$

$$Y^{(i)} = T\left(\left[HA^{(i)}\right]\right), \text{ where } i = 1, 2, \ldots, Q \tag{3}$$

here, $X^{(i)} \in \mathbb{R}^{C \times H \times W}$ represents the i-th feature map of the input, and $HA^{(i)}$ represents the view-specific feature map generated by input $X^{(i)}$ through the hybrid attentive module, where $HA^{(i)} \in \mathbb{R}^{C \times H \times W}$. $Y^{(i)}$ denotes the i-th robust view-specific representation, where $Y^{(i)} \in \mathbb{R}^{N \times d(Y^{(i)})}$.

### Fusion module

The fusion module consists of a fusion network $f(\cdot)$ and a contrastive attentive strategy. $f(\cdot)$ takes this low-dimensional information and utilizes the fully connected layer to make a join then as input to the fusion modules, we obtain the view-common representation G after dimensional scaling and latent space encoding, as follows:

$$G = f(\text{cat}(Y^1, \ldots, Y^Q)), \tag{4}$$

where $G \in \mathbb{R}^{N \times d(G)}$. In order to make G robust, we design a contrastive attentive strategy to avoid destroying the view-specific representations and to maximize the mutual information between G and each Y. The contrastive attentive strategy is discussed in detail below.

### Clustering head C($\cdot$)

The clustering module functions as a guiding task, steering the view common representation G in a good direction. This module leverages existing clustering techniques to assign each data point to a compact space based on G. Consequently, cluster labels can be derived from this process, denoted as labels, which equals C(G).

## Hybrid attentive module

In this subsection, we introduce the hybrid attentive module of detail. The hybrid attentive module is shown in Fig. 3 and contains the channel attention module, the spatial attention module, and positional attention module, and the dimensionality reduction module T() mentioned in the previous section. We first introduce the roles of these modules, before detailing their methods. We go over these three components in the following.

The inner modules of the hybrid attentive module play distinct roles. The channel attention module of the hybrid attentive module enhances feature refinement by generating channel attention maps, aiding in understanding the "content" being learned. Simultaneously, the spatial attention module of the hybrid attentive module determines the "location" of features through the generation of spatial attention maps along the spatial dimension. To mitigate long-distance dependency issues, we adopt the approach of *Hou, Zhou & Feng (2021)*, embedding location features into the channel within the location

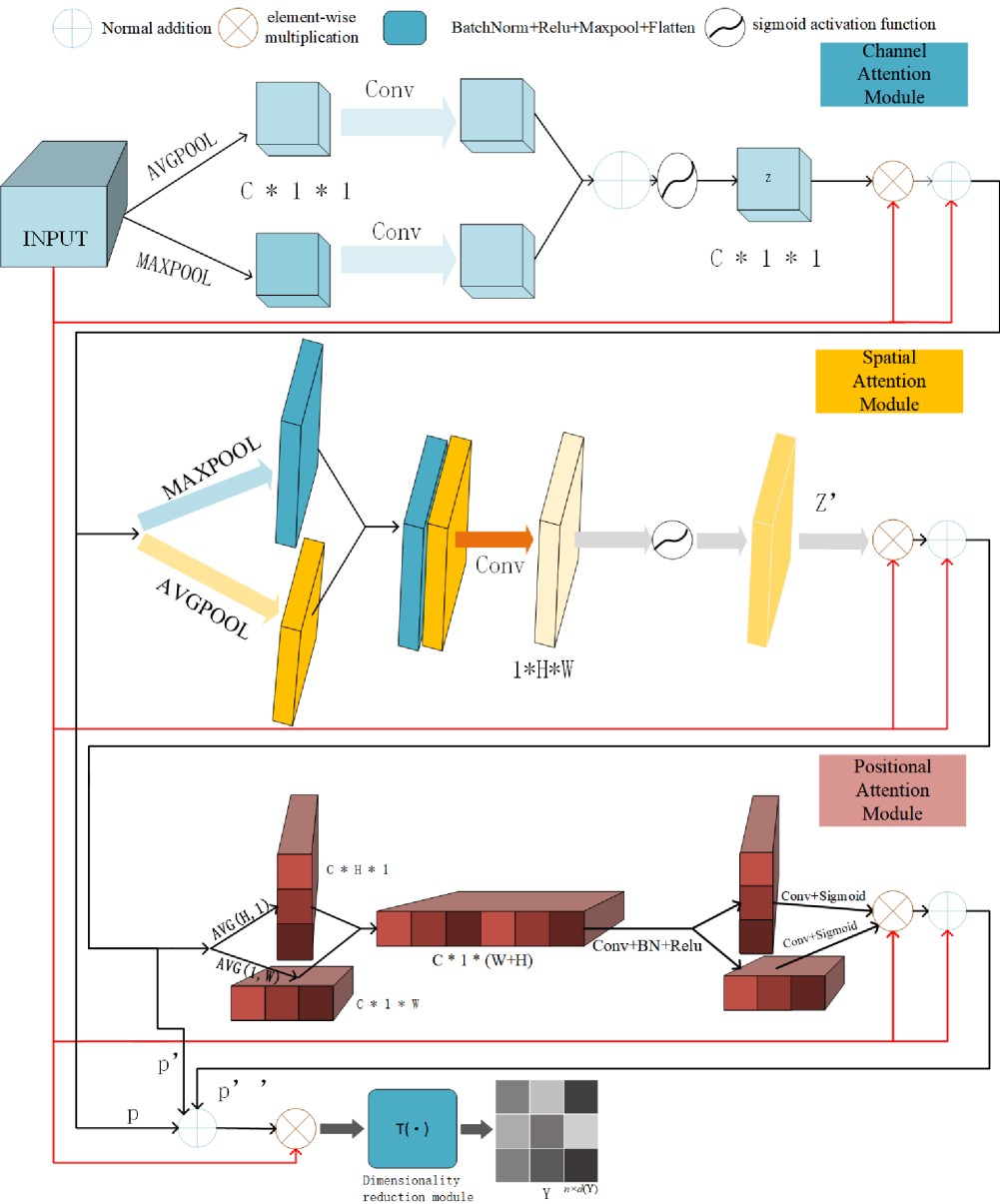

**Figure 3 Illustration of hybrid attentive module workflow.** The input view-specific feature map $\{X^{(i)}{}_{i=1}^{Q}\}$, goes through three rounds of attention map weighting, which includes the channel attention map, spatial attention map, and positional attention map-to produce $\{HA^{(i)}{}_{i=1}^{Q}\}$. Then after T(), the feature map is reducing dimensionality to get the view-specific robust representation $\{Y^{(i)}{}_{i=1}^{Q}\}$.

module. The positional attention module of the hybrid attentive module addresses the long-range dependency problem, ensuring the retention of positional information. Lastly, the dimensionality reduction module converts the 3D feature map $HA \in \mathbb{R}^{C \times H \times W}$ into matrix form $Y \in \mathbb{R}^{N \times d(Y)}$.

The hybrid attentive module enhances downstream tasks by focusing on feature maps with comprehensive channel, spatial, and positional information, thereby obtaining

**Table 1  Summary of key symbols used in this section of the article.**

| Symbols | Explanation |
| --- | --- |
| GAP | Global average pooling |
| GMP | Global maximum pooling |
| Convv$^{i \times i}$ | Convolution of feature maps using a convolution kernel size of i |
| $\sigma$ | Relu activation function |
| $\delta$ | sigmoid activation function |
| BN | Batch Normalization |
| $\otimes$ | multiplied element-by-element |

view-specific robust representations. To prevent convergence to a trivial solution, the hybrid attentive module employs a skip connection and decoupling channel, spatial, and positional information. The overall attention process is summarized in Eq. (5), where $\otimes$ is the element-wise multiplication, $X \in \mathbb{R}^{C \times H \times W}$ is the view-specific feature map, $P \in \mathbb{R}^{C \times H \times W}$ is the channel attention information, $P' \in \mathbb{R}^{C \times H \times W}$ is the spatial attention information, $P'' \in \mathbb{R}^{C \times H \times W}$ is the positional attention information, and $HA \in \mathbb{R}^{C \times H \times W}$ is the refined output.

$$HA = X \bigotimes (P + P' + P'') \tag{5}$$

Table 1 contains pertinent notations for this section. Figure 3 illustrates the entire process, while comprehensive descriptions of each attention module follow.

## Channel attention module

We describe the flow of feature maps in the channel attention module, along with the expression of relevant formulae. Assuming we are initially given a view-specific feature map $X \in \mathbb{R}^{C \times H \times W}$. Following *Woo et al. (2018)*, the channel attention infers a one-dimensional channel attention map $\mathbb{z} \in \mathbb{R}^{C \times 1 \times 1}$ which represents the importance of each channel in the feature map, as shown in Eq. (6).

$$\mathbb{z} = \delta(\mathrm{Conv}^{1 \times 1}(\mathrm{GAP}^C(X)) + \mathrm{Conv}^{1 \times 1}(\mathrm{GMP}^C(X))) \tag{6}$$

$X \in \mathbb{R}^{C \times H \times W}$ possesses a total of C channels, while $\mathbb{z} \in \mathbb{R}^{C \times 1 \times 1}$ signifies a weight for each channel. Therefore, $\mathbb{z} \in \mathbb{R}^{C \times 1 \times 1}$ is a channel weighting information employed to allocate weights to each channel within a network. This empowers the network to prioritize channels with higher weights, thereby augmenting its focus on essential information. Therefore, Let the one-dimensional channel attention graph $\mathbb{z} \in \mathbb{R}^{C \times 1 \times 1}$ be corrected for the feature map $X \in \mathbb{R}^{C \times H \times W}$ by correcting it with the feature map $\mathbb{z}_c \in \mathbb{R}^{C \times H \times W}$, *i.e.,* letting the two be multiplied element-by-element to obtain a channel attention graph-weighted feature map $\mathbb{z}_c \in \mathbb{R}^{C \times H \times W}$, as follows in Equation (7). The channel attention module accentuates the significance of particular channels in the overall computation process.

$$\mathbb{z}_c = (\mathbb{z} \otimes X). \tag{7}$$

However, previous work (*He et al., 2016*) shows that as the network becomes deeper (*i.e.,* the number of layers increases), the network may learn mundane solutions. For this

reason, we introduce skip connections. As a result, the channel attention module of the hybrid attentive module produces the feature graph information $P \in \mathbb{R}^{C \times H \times W}$, as shown in Eq. (8).

$$P = X + \mathbb{z}_c \tag{8}$$

## Spatial attention module

We detail the flow of feature maps in the spatial attention module as well as the expression of formulas. We initiate the process by applying both the GAP and GMP operations to the weighted feature maps $P \in \mathbb{R}^{C \times H \times W}$ derived from the channel attention maps, performed along the channel dimensions. This yields the feature description operators, known as $GAP^s$, another called $GMP^s$. Subsequently, we merge $GAP^s$ and $GMP^s$, and apply a $7 \times 7$ convolution operation to reduce dimensionality. Next, we determine the direction of the two-dimensional spatial attention map $\mathbb{z}' \in \mathbb{R}^{1 \times H \times W}$ by applying the sigmoid activation function, which plays a crucial role in highlighting relevant features, as shown in Eq. (9).

$$\mathbb{z}' = \delta(Conv^{7 \times 7}(GAP^s(P); GMP^s(P))) \tag{9}$$

$\mathbb{z}' \in \mathbb{R}^{1 \times H \times W}$ is a 2D weight map that records the magnitudes of location weights within the process. This facilitates the network in attending to regions with prominent features. The large weights of the 2D maps represent stronger features. Following *Woo et al. (2018)*, we rectify the 2D spatial attention map, $\mathbb{z}' \in \mathbb{R}^{1 \times H \times W}$, by correcting it with $P \in \mathbb{R}^{C \times H \times W}$ through the channel attention module. In simple terms, these two entities undergo element-wise multiplication, leading to the creation of a channel attention map-weighted feature map denoted as $\mathbb{z}_s \in \mathbb{R}^{C \times H \times W}$, as shown in Eq. (10).

$$\mathbb{z}_s = \mathbb{z}' \otimes P \tag{10}$$

We add the above-generated attention information to the original view-specific feature map X. This is to prevent the attention information from focusing on "biased" information, so the feature map information $P' \in \mathbb{R}^{C \times H \times W}$ generated by the channel attention module of the hybrid attentive module is as shown in Eq. (11).

$$P' = X + \mathbb{z}_s \tag{11}$$

## Positional attention module

The following is a detailed description of the workflow and formula expression of the positional attention module. We put $P' \in \mathbb{R}^{C \times H \times W}$, pooling along two different directions of the feature map with two pooling kernels $(H, 1)$ and $(1, W)$ yields two high direction embedded information feature maps $I_c^h \in \mathbb{R}^{C \times H \times 1}$, and wide direction embedded information feature maps embedded information feature map $I_c^w \in \mathbb{R}^{C \times 1 \times W}$, respectively. Therefore, the output of the c-th channel at height h is expressed as in Eq. (12).

$$I_c^h(h) = \frac{1}{w} \sum_{i=0}^{W} P'(h, i). \tag{12}$$

Similarly, the output of the c-th channel with width w is expressed as in Eq. (13).

$$I_c^w(w) = \frac{1}{H}\sum_{j=0}^{H} P'(w, j). \tag{13}$$

To establish cross-channel links, we concatenate the two embedded information feature maps obtained along the spatial dimension to obtain comprehensive global information. Subsequently, a ReLU activation follows a $1 \times 1$ convolutional transformation, resulting in $f_z \in \mathbb{R}^{C \times 1 \times (W+H)}$, as illustrated in Eq. (14).

$$f_z = \sigma(\text{Conv}^{1\times1}(I_c^h(h); I_c^w(w))) \tag{14}$$

This is done to subsequently perform a split operation along the spatial dimension to obtain two separated feature maps $f^h \in \mathbb{R}^{C \times H \times 1}$, and $f^w \in \mathbb{R}^{C \times 1 \times W}$, as shown in Eq. (15).

$$f^h, f^w = \text{split}(f_z). \tag{15}$$

Then, the $1 \times 1$ convolution variant is sigmoid activated to obtain the positional information feature maps located on the width and the height $z_h \in \mathbb{R}^{C \times H \times 1}$ and $z_w \in \mathbb{R}^{C \times 1 \times W}$. Its formula is obtained as:

$$z_h = \delta(\text{Conv}_h^{1\times1}(f^h)), \tag{16}$$
$$z_w = \delta(\text{Conv}_w^{1\times1}(f^w)). \tag{17}$$

Finally, let the positional weighting information $z_h$ and $z_w$ be corrected after dot multiplication with the feature map $P' \in \mathbb{R}^{C \times H \times W}$. That is, multiply them element-wise to obtain positional weighting information with the positional attention information of the embedded channel $z_p(i, j) \in \mathbb{R}^{C \times H \times W}$, as shown in Eq. (18).

$$z_p(i, j) = z_h(j) \otimes z_w(i) \otimes P' \tag{18}$$

To prevent the positional attention information from focusing on "biased" information, we add to the positional attention information the original view-specific feature map $X \in \mathbb{R}^{C \times H \times W}$. Thus, the feature map information $P'' \in \mathbb{R}^{C \times H \times W}$, produced by the positional attention module of the hybrid attentive module, as shown in Eq. (19).

$$P'' = X + z_p(i, j) \tag{19}$$

All in all, our designed hybrid attentive module efficiently allocates effective feature weights, and we summarize its effectiveness in two aspects. (1) The produced weighting information, $z \in \mathbb{R}^{C \times 1 \times 1}$, $z' \in \mathbb{R}^{1 \times H \times W}$, $z_h \in \mathbb{R}^{C \times H \times 1}$, $z_w \in \mathbb{R}^{C \times 1 \times W}$, capture comprehensive attention to feature maps from multiple perspectives: channel dimension (c-th), spatial dimensions (h and w-th), and high and wide positional dimensions (c, h-th and c, w-th). This differs from previous attention mechanisms that focused on one or two dimensions. We exhaustively verify in the ablation experiments section that features map weighting that considers full aspects will be better than one that considers only one or two dimensions. (2) Simultaneously, a crucial design aspect of the hybrid attention mechanism is the addition of the original feature map X after generating each attention map. For

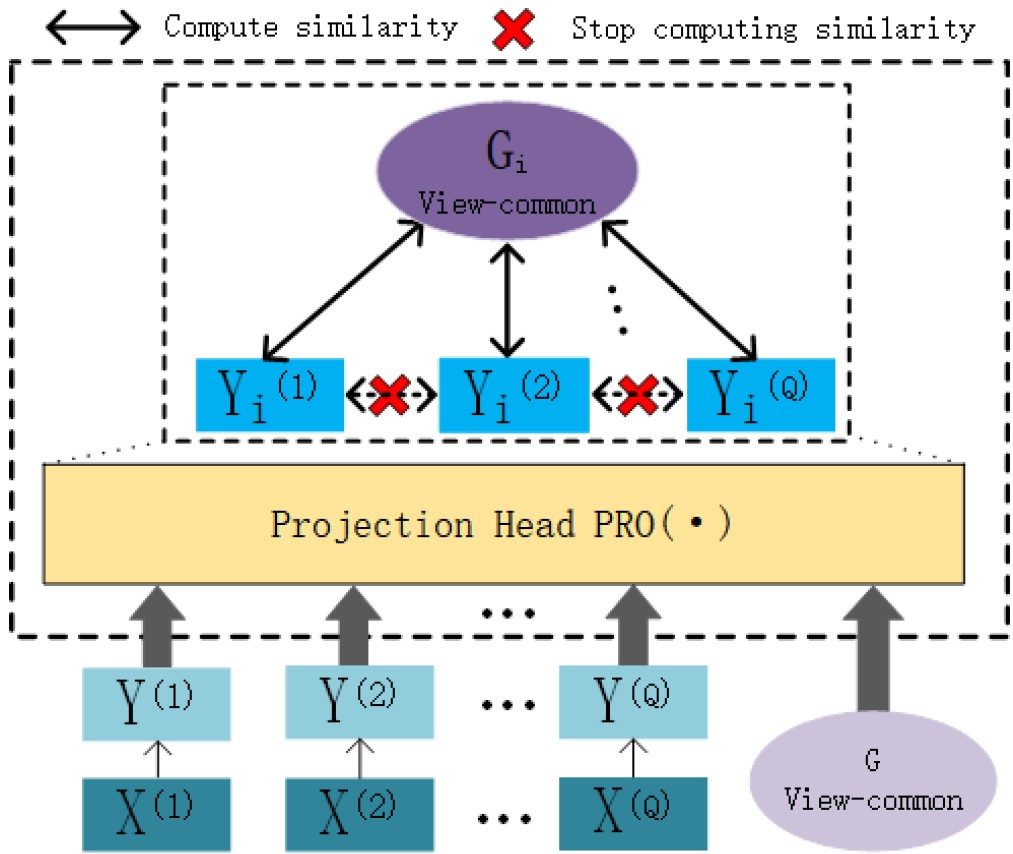

**Figure 4** **Illustration of the workflow of the contrastive attentive strategy.**

instance, assuming $z \in \mathbb{R}^{3 \times 1 \times 1}$ with weight distribution 0, 0.9, 0.1, where the channel with weight 0 is significant, we add the original feature map to correct biased information. This design ensures correction even when attention weights deviate towards incorrect information. Consequently, we attribute the success of the hybrid attentive module to these two design elements.

## Contrastive attentive strategy

In this chapter, we present the contrastive attentive strategy (COATS), as illustrated in Fig. 4. COATS processes the input view-specific robust representations and the view-common representation by passing them through the mapping head $PRO(\cdot)$. Subsequently, it computes the cosine similarity between the two representations and calculates the contrastive loss based on this cosine similarity. It's worth noting that COATS does not directly align view-specific representations with each other, which helps in preserving the completeness of the views. The mapping head $PRO(\cdot)$ follows the same settings as previously defined in our work (*Chen et al., 2020b*) for constructing the projection header.

We conclude that the contrastive attentive strategy has two advantages: (1) it ensures that the intrinsic structure of the original view is not destroyed, and (2) it weights the view-specific representations to obtain a robust view-specific representation and thus

a more comprehensive view-common representation. Following *Chen et al. (2020b)*, we compute the similarity between view-specific robust representations (after projection head PRO($\cdot$)) and the view-common representation of the view (after projection head PRO($\cdot$)) as cosine similarity, as shown in Eq. (20).

$$d_{i,j} = \frac{(G_i)^{\top}(Y_j^{(q)})}{\| G_i \| \cdot \| Y_j^{(q)} \|} \tag{20}$$

In this context, the notation $d_{i,j}(i \neq j)$ denotes the calculation of similarity for a pair of negative examples, while $d_{i,i}$ signifies the computation of similarity for a pair of positive examples. Intuitively, $d_{i,j}$ means that at this point the view-common representation $G_i$ computes the similarity with the view-specific robust representations of the negative examples $Y_j^{(q)}$, and vice versa. Following *Chen et al. (2020b)*, we design a contrastive learning loss function for AMCFCN, as shown in the following:

$$L_{sim}^{AMCFCN} = -\sum_{i=1}^{N} \log \frac{\exp\left(\frac{d_{i,i}}{\tau}\right)}{\sum_{j}^{2N} \lessdot_{[i \neq j]} \exp\left(\frac{d_{i,j}}{\tau}\right)}, \tag{21}$$

$$L_c = \sum_{q=1}^{Q} L_{sim}^{AMCFCN}. \tag{22}$$

The exp() denotes an exponential function with a natural number e as the base. The symbol N represents the number of samples, while $(i, i)$ and $(i, j)$ represent a pair of positive and a pair of negative examples, respectively; $\lessdot_{[i \neq j]} \in \{0, 1\}$ denotes an indicator function that takes the value of one if $i = j$, and $\tau$ denotes a temperature constant. In our experiments, we adopt the optimal setting of $\tau = 0.1$ for the temperature parameter suggested by *Ke et al. (2021)*.

## The objective function

The total loss function is used to measure the degree of difference between the predicted and true values of the model. Before this, the whole network process lacked a certain steering direction, and we chose the clustering steering instead of the task, and we empirically used the off-the-shelf deep divergence-based DDC (*Kampffmeyer et al., 2019*) online clustering task. It assigns the consistency representation G to a compact space, *i.e.*, a soft label can be obtained from this go, *i.e.*, B = C(G). To reduce the computational cost, AMCFCN only uses DDC to constrain the view-common representation G.

The loss function of DDC is described below. DDC uses Cauchy–Schwarz scatter (CS-scatter) to enhance the separability and compactness of the clustering. We predefine some theoretical foundations, where B is an n × k cluster assignment matrix and $b_i$ is the *i*-th column of matrix B, and $\varrho$ is Gaussian kernel bandwidth, defaulting 0.15. $\mathbb{K}$ is computed as in Eq. (23). $m_{i,j}$ is computed as in Eq. (24), where $e_j$ is corner *j* of the standard simplex on $\mathbb{R}^k$.

$$\mathbb{K}_{i,j} = \exp(- \| G_i - G_j \|^2 / (2\varrho)^2) \tag{23}$$

$$m_{i,j} = \exp(- \| b_i - e_j \|^2) \tag{24}$$

triu(BB)$^T$ denotes B$^T$ the upper strictly triangular element, and $\mathbb{K}$ denotes kernel similarity matrix computed by $\mathbb{K}_{i,j}$. Then the loss function of DDC consists of three terms as shown below:

$$L_{ddc} = \sum_{i=1}^{k-1}\sum_{j>i}^{k} \frac{b_i^T \mathbb{K} b_j}{\sqrt{b_i^T \mathbb{K} b_j b_j^T \mathbb{K} b_i}} + \sum_{i=1}^{k-1}\sum_{j>i}^{k} \frac{m_i^T \mathbb{K} m_j}{\sqrt{m_i^T \mathbb{K} m_j m_j^T \mathbb{K} m_i}} + \text{triu}(BB)^T \qquad (25)$$

In summary, we combined the objective function of AMCFCN as follows:

$$L_{sum} = L_{ddc} + \vartheta L_c, \qquad (26)$$

here, $\vartheta$ is a trade-off parameter intended to restrict the contrastive loss term from becoming excessively large. $\vartheta$ is meticulously determined in the experimental section.

# EXPERIMENTS AND RESULT ANALYSIS

In this section, we initially introduce the datasets: E-MNIST, E-FMNIST, COIL-20, COIL-100, and the experimental environment. Subsequently, we account for the model Performance. Then, we conduct ablation experiments on two key methods: the hybrid attentive module and the contrastive attentive strategy. Additionally, we explore the hyperparametric analysis of the loss function. Lastly, we introduce convergence of model analysis.

## Dataset

We evaluate the model AMCFCN on four public multi-view datasets and compare it to seven other mainstream models. The four datasets are.

(1) Edge-MNIST (E-MNIST) (*Liu & Tuzel, 2016*): which is from the staff of the US Census Bureau. This is a large and widely used benchmark dataset. A set of public benchmark datasets consists of 60,000 handwritten digital images (10 categories) of $28 \times 28$ pixels. In this dataset, there are two views containing the original figure and the edge detection versions, respectively.

(2) Edge-Fashion MNIST (E-FMNIST) (*Xiao, Rasul & Vollgraf, 2017*): This is provided by Zalando's research department (German fashion tech company). This dataset, characterized by its diverse range of clothing items in terms of shapes, colors, and patterns, presents a higher level of complexity compared to the standard MNIST dataset. We create a second view by running the same edge detector used to create Edge-MNIST.

(3) COIL-20 (*Nene, Nayar & Murase, 0000*): This is Columbia University's description of grayscale images containing 20 categories from separate perspectives. We create a three-view dataset by randomly grouping the coil-20 dataset into three sets. The details of creating the three views are covered in the next paragraph.

(4) COIL-100 (*Geusebroek, Burghouts & Smeulders, 2005*): This is from Columbia University; separate RGB images containing 100 categories were taken from different angles, and we created a three-view dataset by randomly grouping the coil-100 dataset into three sets. The coil-100 constructs the three views in the same way as the coil-20.

**Table 2  Overview of the dataset.**

| Data set | Samples | Categories | View | Dimensionality |
|---|---|---|---|---|
| E-MNIST | 60,000 | 10 | 2 | $1 \times 28 \times 28$ |
| E-FMNIST | 60,000 | 10 | 2 | $1 \times 28 \times 28$ |
| Coil-20 | 480 | 20 | 3 | $1 \times 28 \times 28$ |
| Coil-100 | 2,400 | 100 | 3 | $3 \times 28 \times 28$ |

We randomize the objects of a category using a random arrangement technique to generate a random index for image generation, the purpose of utilizing random is for the diversity of the selected view images. Multiple views are constructed from the generated indexes. For example, in Coil-20, a category comprises 72 images, and when three views are constructed, the index can form a $3 \times 24$ matrix. The selected images are organized into three views and stored as an array that represents three different views of the images. In short, items for coil-20 are generated by randomly selecting images related to the project and organizing them into three different views.

The details are shown in Table 2:

## Implementation details

The experimental system runs on Ubuntu 18.04.6, utilizing hardware comprising an NVIDIA RTX 3090 GPU with 16GB of memory and an AMD EPYC 7702 64-core Processor. The software includes CUDA 11.4 and Python 3.9.7, while the deep learning framework PyTorch (*Paszke et al., 2019*) is employed for model development and validation on a publicly available benchmark dataset.

The model is validated on a publicly available benchmark dataset. For the MNIST class dataset, key hyperparameters are configured as follows: the training batch epoch is set to 100; model optimization employs the ADAM optimizer with default parameters; the batch size is fixed at 100; and reference to specific encoder CNN structures and configurations in *Abavisani & Patel (2018)*. The training dataset's batch size is 128; the initial learning rate is 0.001; and the hidden representation dimensionality is 288. On the other hand, for the COIL class dataset, we adopt AlexNet (*Krizhevsky, Sutskever & Hinton, 2017*) as the view-specific encoder, with a batch size of 24, an initial learning rate of 0.0001, and a hidden representation size of 1,024.

## Model performance

In this subsection, we conduct a series of experiments to evaluate the performance of the model, and we explain our experimental design and results. We introduce model evaluation metrics, model stability experiments, model visualization experiments, and model comparison experiments.

### Evaluation metrics

Two well-known metrics are considered in the comparison experiments, they are accuracy (ACC) and normalized mutual information (NMI). The ACC represents the total proportion of clustered samples to the total samples, while the NMI is based on the information theory idea and is used to measure the similarity between two groups of

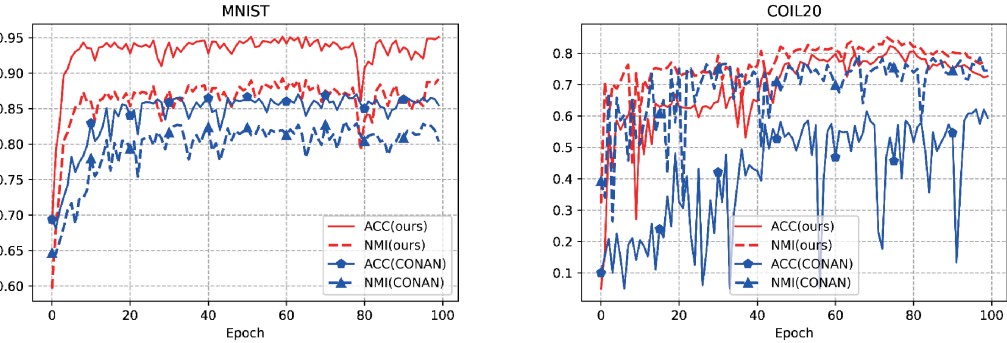

**Figure 5** Stability performance of our model and the next best model on two public datasets.

samples. The larger value of the two sets of evaluation indexes represents the better clustering effect. The predicted clustering labels are denoted as $F_j$, where $F_j \in F$ denotes the predicted clustering label; the true label is denoted as $D_j$, $D_j \in D$ denotes the true label of the dataset; the map() is the mapping function for the optimal one-to-one assignment of clusters to labels (*Kuhn, 1955*). Among them, $I(D_j, map(F_j))$ is an indicator function, when $D_j = map(F_j)$ the indicator function is 1, otherwise it is 0; then the ACC metric is defined as:

$$ACC = \frac{\sum_{j=1}^{N} I(D_j, map(F_j))}{N}. \tag{27}$$

$I(D; F)$ and $H(\cdot)$ denote the true and predicted clustered label mutual information, respectively, and the entropy function. The NMI metric is defined as shown in (28).

$$NMI = \frac{2I(D; F)}{(H(D) + H(F))} \tag{28}$$

### Stability experiments

In the field of unsupervised learning, stability plays a crucial role in its performance and reliability. The stabilization of the clustering is indirectly illustrated by the evolution of ACC and NMI. The stability evaluation experiments for AMCFCN are conducted by monitoring the rate of change in these two metrics over one run. AMCFCN and the next best model (CONAN) are executed on two datasets, there are E-MNIST and COIL-20, from which the rate of change of ACC and NMI metrics is obtained, as follows in Fig. 5.

As shown in Fig. 5, the rate of change of the two metrics of CONAN is very large, which indirectly indicates that the performance of CONAN is not stable enough. Compared to CONAN, AMCFCN exhibits elevated levels in both metrics with minimal rates of change. It is noteworthy that the contrastive attentive strategy yields a robust view-common representation, contributing to the stability demonstrated by AMCFCN.

### Visualization experiments

To show the effectiveness of the contrastive attentive strategy, we visualize the view-common representation. We demonstrate the clustering process on the E-MNIST and

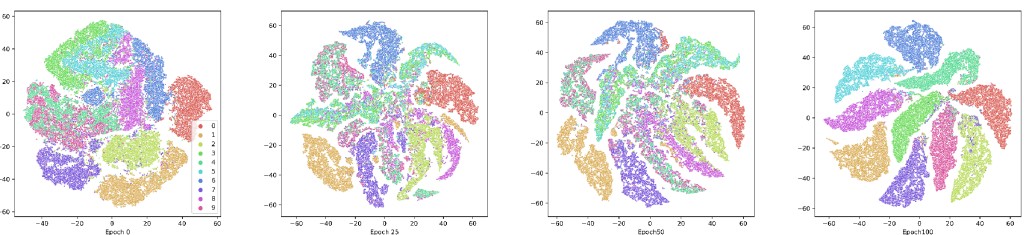

**Figure 6** Visualization of the view-common representation using T-SNE on E-MNIST dataset at the 1st, 25th, 50th, and 100th training.

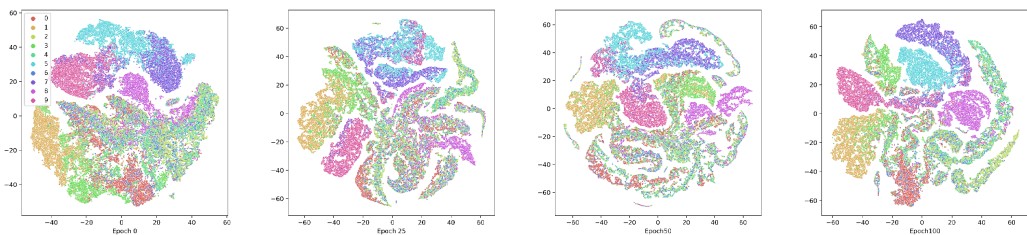

**Figure 7** Visualization of the view-common representation using T-SNE on FMNIST dataset at the 1st, 25th, 50th, and 100th training.

COIL-20 datasets, as depicted in Figs. 6 and 7. The following T-SNE (*Vander Maaten & Hinton, 2008*), is a dimensionality reduction technique that visualizes high-dimensional data by preserving pairwise similarities in a lower-dimensional space. AMCFCN employs T-SNE to project the view-common representation onto the 2D plane for both the E-MNIST and E-FMNIST datasets. The visualization reveals that the structure specific to clustering becomes more compact and separable as the training period is extended. The two experimental studies demonstrate that the view-specific robust representation obtained through the hybrid attentive module enhances the view-common representation through view-specific representation fusion, which in turn improves the stability of the multi-view clustering.

### Comparison experiments

AMCFCN evaluates its performance against seven current mainstream multi-view clustering models to assess its effectiveness. The comparison models include:

  I.  Deep Canonical Correlation Analysis (DCCA) (*Gao et al., 2020*);
 II.  Deep Canonically Correlated Auto-encoders (DCCAE) (*Wang et al., 2015*);
III.  Deep Multi-modal Subspace Clustering (DMSC) (*Yang et al., 2021*);
 IV.  Deep Adversarial Multi-view Clustering (DAMC) (*Li et al., 2019*);
  V.  End-to-end Adversarial attention network for Multi-modal Clustering (EAMC) (*Zhou & Shen, 2020*);
 VI.  Autoencoder in autoencoder networks (Ae$^2$-Nets) (*Zhang, Liu & Fu, 2019*);
VII.  Contrastive Fusion Networks for Multi-view Clustering (CONAN) (*Ke et al., 2021*);

**Table 3  Comparison results of different data sets in different models (%).**

| Data set | E-MNIST | | Coil-20 | | E-FMNIST | | Coil-100 | |
| --- | --- | --- | --- | --- | --- | --- | --- | --- |
| Evaluation indicators | ACC | NMI | ACC | NMI | ACC | NMI | ACC | NMI |
| DCCA (*Gao et al., 2020*) | 47.6 | 44.3 | | | 42.6 | 41.9 | | |
| DCCAE (*Wang et al., 2015*) | 50.8 | 47.5 | | | 45.8 | 43.6 | | |
| DMSC (*Yang et al., 2021*) | 65.3 | 61.4 | 65.1 | 72.0 | 52.4 | 51.8 | 54.1 | 53.8 |
| DAMC (*Li et al., 2019*) | 64.6 | 59.4 | 67.2 | 72.9 | 49.5 | 48.7 | 52.6 | 70.1 |
| Ae$^2$-Nets (*Zhang, Liu & Fu, 2019*) | 53.7 | 46.3 | 71.7 | 81.2 | 41.0 | 39.2 | 54.1 | 75.3 |
| EAMC (*Zhou & Shen, 2020*) | 66.8 | 62.8 | 69.0 | 75.3 | 54.1 | **62.2** | 51.8 | 68.0 |
| CONAN (*Ke et al., 2021*) | 90.5 | 86.1 | 72.5 | 80.3 | 59.2 | 55.2 | 55.6 | 77.4 |
| AMCFCN (OURS) | **95.1** | **89.2** | **82.2** | **85.2** | **63.2** | 55.8 | **66.4** | **84.8** |

**Notes.**

Large-scale or high-dimensional datasets require substantial computational resources. DCCA and DCCAE are applicable exclusively to two-view datasets. Bolded values represent the highest value of an indicator for this experiment. 'p' denotes the inclusion of the current module, while '-' signifies its exclusion.

To mitigate the influence of randomness in experimental outcomes, AMCFCN conducts five repetitions of training on each dataset and utilizes the average values for evaluation. We conduct a comparative experiment involving seven classical deep models, evaluating AMCFCN alongside the baseline model across four datasets. The resulting clustering performance, expressed in percentages (%), is presented in Table 3. As evident from the comparative results in the table, AMCFCN outperforms the suboptimal model on the Coil-20 and E-MNIST datasets across two metrics, achieving improvements of 10% and 4.9%, another 4.6% and 3.1%, respectively in ACC and NMI.

To make it easier to observe AMCFCN, we visualize the seven models and AMCFCN in Fig. 8. AMCFCN is represented in violet. Higher bars in the figure represent better performance, and it can be seen from the figure that the bar levels of AMCFCN are basically higher than those of the seven methods. Thus, AMCFCN outperforms the mainstream methods on most datasets, but its NMI metrics on the E-FMNIST dataset are slightly lower than those achieved by EAMC.

## Ablation experiments

In this subsection, we conduct a total of two ablation experiments. The first ablation experiments involve the hybrid attentive module proposed in the article, with the ablation focus on the three types of attention within the module. The other ablation experiments assess the contrastive attentive strategy's impact when contrastive learning and attention mechanisms are absent in AMCFCN. In the experimental data table below, '$\sqrt{}$' denotes the inclusion of the current module, while '-' signifies its exclusion.

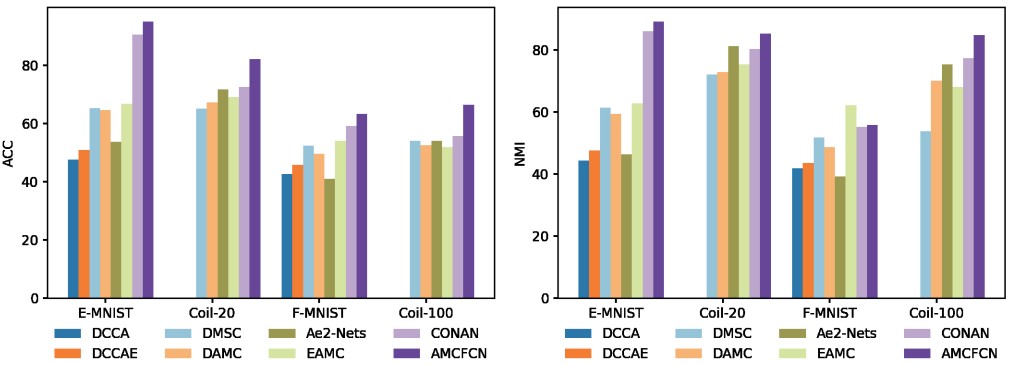

**Figure 8    Performance comparison of AMCFCN with seven multi-view clustering models.**

**Table 4    Internal ablation study of the hybrid attentive module on EMNIST dataset.**

| Channel attention module | Spatial attention module | Positional attention module | ACC (%) | NMI (%) |
| --- | --- | --- | --- | --- |
| √ | √ | √ | **95.1** | **89.2** |
| √ | – | – | 86.5 | 84 |
| √ | – | √ | 84.0 | 79.7 |
| – | √ | – | 87.5 | 84.6 |
| – | √ | √ | 82.1 | 75.5 |
| √ | √ | – | 87.9 | 83.8 |

**Notes.**
Bolded values represent the highest value of an indicator for this experiment. '√' denotes the inclusion of the current module, while '–' signifies its exclusion.

The hybrid attentive module for AMCFCN Ablation experiments conducts ablation experiments on the EMNIST dataset, as detailed in Table 4. The results demonstrate that the hybrid attentive module internally achieves the best performance when all three attention modules are employed, with metrics of 95.1% and 89.2%, respectively. In contrast, the hybrid attentive module performed worst when relying only on the spatial and positional attention modules, with metrics of 82.1% and 75.5%, respectively. These ablation experiments demonstrate the robust noise suppression capability of the hybrid attentive module, especially when using three modules, which allows the hybrid attentive module to focus more comprehensively on the feature information of the feature map. This enhancement allows AMCFCN to improve its information capture, thus allowing the clustering to obtain a good result.

This ablation experiment aims to assess the contrastive attentive strategy. The experiment involves using the contrastive attentive strategy to obtain a view-common representation, followed by incomplete strategies in subsequent experiments. These incomplete strategies include one utilizing only the contrastive learning strategy, another employing the attention mechanism strategy, and the last one employing no specific strategy. To validate the superior performance of the proposed fusion strategy of the contrastive attentive strategy,

**Table 5 Contrastive attentive strategy ablation findings on four datasets.**

|  | Hybrid attentive module | Contrastive learning | ACC (%) | NMI (%) |
|---|---|---|---|---|
| E-MNIST | √ | √ | **95.1** | **89.2** |
|  | √ | – | 87.4 | 85.0 |
|  | – | √ | 87.7 | 84.4 |
|  | – | – | 87.3 | 84.6 |
| FMNIST | √ | √ | **63.2** | **55.5** |
|  | √ | – | 61.4 | 55.4 |
|  | – | √ | 59.6 | 54.4 |
|  | – | – | 61.4 | 53.8 |
| Coil-20 | √ | √ | **82.2** | **85.2** |
|  | √ | – | 70.2 | 76.4 |
|  | – | √ | 62.7 | 71.5 |
|  | – | – | 69.7 | 78.1 |
| Coil-100 | √ | √ | **66.4** | **84.8** |
|  | √ | – | 53.8 | 79.5 |
|  | – | √ | 36.3 | 65.6 |
|  | – | – | 38.9 | 67.4 |

**Notes.**

Bolded values represent the highest value of an indicator for this experiment. '√' denotes the inclusion of the current module, while '–' signifies its exclusion.

ablation experiments conduct on four datasets, as detailed in Table 5. When employing the contrastive attentive strategy, AMCFCN consistently outperforms the incomplete strategies across both metrics on the various datasets. The improvement is particularly noteworthy on the COIL-100 dataset, where the use of the contrastive attentive strategy leads to a 27.5% increase in ACC and a 17.4% increase in NMI compared to the no-strategy approach.

We compare the differences in model learning performance under each of the four steps, as shown in Fig. 9. Specifically, Step-A is the model with the hybrid attentive module, Step-B is the model with contrastive learning, and Step-C is the model with none of the why strategy. In Fig. 9, where higher bars denote better performance, it is evident that AMCFCN achieves the most favorable results when utilizing COATS. The utilization of the hybrid attentive strategy enhances AMCFCN's performance compared to models employing incomplete strategies (Step-A, Step-B) or no strategies (Step-C) across all four datasets.

## Hyperparametric analysis of the objective function

We conduct a hyperparametric analysis experimental study for $\vartheta$ of Eq. (26). $\vartheta$ represents the hyperparameter for the contrastive loss within the objective function, and we analyze the study on the E-MNIST dataset, utilizing the optimal value determined for $\vartheta$ as the parameter value for all experiments in this context. Then, AMCFCN is analyzed and studied on the EMNIST dataset, employing values ranging from $10^{-3}$ to 10. The specific experimental results are displayed in Table 6.


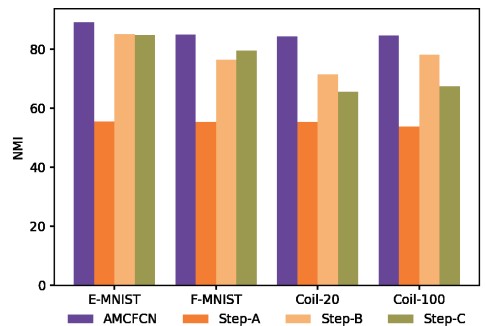

**Figure 9** Ablation experiments are performed on four multi-view datasets.

**Table 6  Impact of hyperparameter on AMCFCN model.**

| Dataset | The size of the $\vartheta$ | ACC (%) | NMI (%) | BEST_LOSS |
|---------|------------------------------|---------|---------|-----------|
| E-MNIST | 10 | 86.7 | 80.1 | 184.4 |
|         | 1 | 87.1 | 83.6 | 19.0 |
|         | 0.1 | 87.7 | 84.5 | 2.47 |
|         | 0.01 | 95.1 | 89.2 | 0.82 |
|         | 0.001 | 87.4 | 85.4 | 0.64 |

It is observed that when $\vartheta$ is set to 10, the ACC and NMI are 86.7% and 80.1%, respectively, resulting in the best total loss value of AMCFCN reaching 184.4. Conversely, when $\vartheta$ is set to 0.001, the ACC and NMI are 87.4% and 85.4%, respectively, with a total loss of only 0.64. Values of $\vartheta$ that are excessively large or small can cause one of the loss components to dominate network training, contrary to our intended hyperparameter setting. With $\vartheta$ set to 0.01, the ACC and NMI are 95.1% and 85.4%, respectively, while the total loss is 0.82. Therefore, we consider $\vartheta = 0.01$ an appropriate value, ensuring that the contrastive loss and clustering loss of AMCFCN are of comparable magnitude, allowing both loss components to guide network training effectively.

## Convergence analysis

In this subsection, we focus on the convergence of the methods. Some of the popular methods based on autoencoders, such as COMPLETER (*Lin et al., 2021*) and EMC-Nets (*Ke et al., 2022*), require 200 and 400 training epochs, respectively, to achieve good convergence. We discuss the convergence of AMCFCN by visualizing the change of loss values with training epochs, and the convergence analysis of AMCFCN loss values on four datasets is shown in Fig. 10.

The visualization analysis in Fig. 10 reveals that the loss value of AMCFCN consistently decreases with each epoch, ultimately converging to a stable target value by the 30th training epoch. The rapid and effective convergence of AMCFCN is attributed to the COATS. This strategy preserves the complete view-specific representations, eliminating the necessity to continually learn features from corrupted view-specific representations

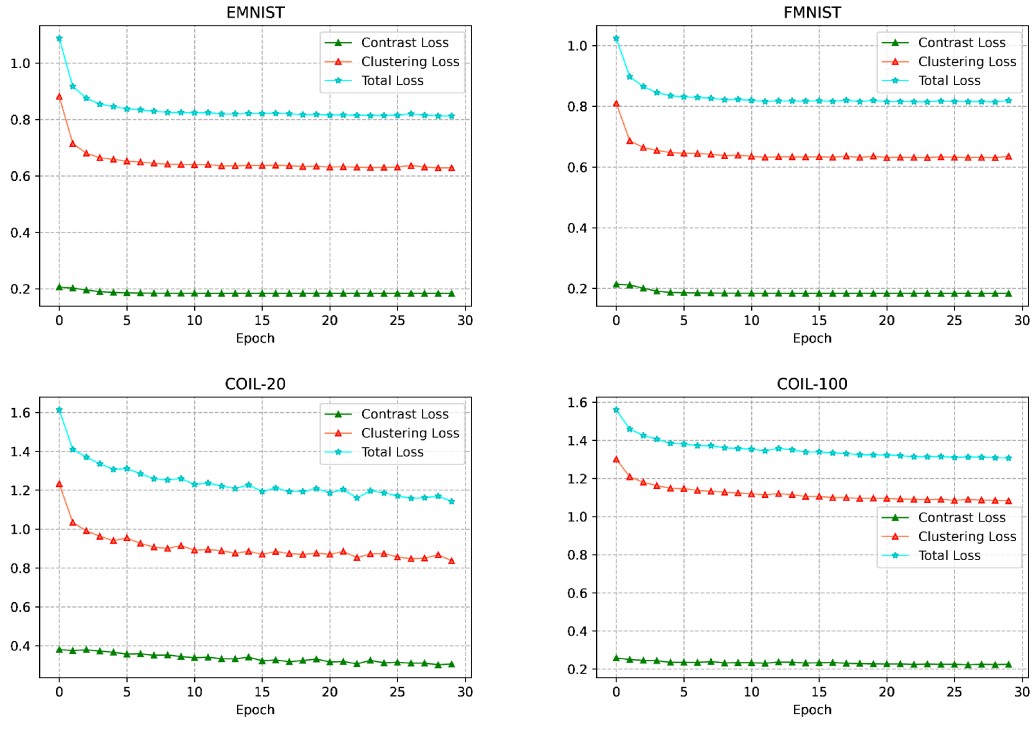

**Figure 10  Change in loss values with a training period.**

over time. In summary, the accelerated convergence of AMCFCN can be attributed to a contrastive attentive strategy. The strategy retains both view completeness and improves the robustness of the view-common representation, thus expediting the convergence process. AMCFCN requires about 30 training epochs to achieve convergence on the four datasets, which is much faster than the autoencoder-based methods to converge.

## CONCLUSIONS

In this work, we introduce a novel framework called AMCFCN, validating its effectiveness on four multi-view datasets, and demonstrating a significant enhancement in unsupervised representations. However, it is worth noting that AMCFCN shows a relatively smaller improvement compared to EAMC (*Zhou & Shen, 2020*) on the E-FMNIST dataset. This observation sheds light on a potential limitation of AMCFCN within the domain of multi-view clustering. This is because the study primarily utilizes smaller or medium-sized datasets, and the AMCNCF does not take into account the balance between contrastive learning method and attention mechanism. Moving forward, our future research will focus on extending AMCFCN's suitability to very large datasets, and aim to strike an optimal balance between attention mechanism and contrastive learning methods.

### Funding

This work was supported by the National Natural Science Foundation of China (No. 52305550), the Guangdong University Scientific Research Project, China (No. 2018WZDXM014) and the Joint Research and Development Fund of Wuyi University and Hong Kong and Macau (2019WGALH21). The funders had no role in study design, data collection and analysis, decision to publish, or preparation of the manuscript.

### Grant Disclosures

The following grant information was disclosed by the authors:
National Natural Science Foundation of China: 52305550.
Guangdong University Scientific Research Project, China: 2018WZDXM014.
Joint Research and Development Fund of Wuyi University and Hong Kong and Macau: 2019WGALH21.

### Competing Interests

The authors declare there are no competing interests.

### Author Contributions

- Huarun Xiao conceived and designed the experiments, performed the experiments, analyzed the data, performed the computation work, prepared figures and/or tables, authored or reviewed drafts of the article, and approved the final draft.
- Zhiyong Hong conceived and designed the experiments, analyzed the data, authored or reviewed drafts of the article, and approved the final draft.
- Liping Xiong conceived and designed the experiments, analyzed the data, authored or reviewed drafts of the article, and approved the final draft.
- Zhiqiang Zeng conceived and designed the experiments, analyzed the data, authored or reviewed drafts of the article, and approved the final draft.

### Data Availability

The data is available at figshare: xiao, huarun (2023). Journal_AMCFCN. figshare. Journal contribution. Available at https://doi.org/10.6084/m9.figshare.24525220.v1.

### Supplemental Information

Supplemental information for this article can be found online at http://dx.doi.org/10.7717/peerj-cs.1906#supplemental-information.

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
