# Peer review of "AMCFCN: attentive multi-view contrastive fusion clustering net"

_PeerJ Computer Science, doi:10.7717/peerj-cs.1906_

## Round 0.1 · original submission · Major Revisions

The work is interesting. Please revise it according to the comments. Then it will be evaluated again.

**Language Note:** The review process has identified that the English language must be improved. PeerJ can provide language editing services - please contact us at copyediting@peerj.com for pricing (be sure to provide your manuscript number and title). Alternatively, you should make your own arrangements to improve the language quality and provide details in your response letter. – PeerJ Staff

Reviewer 1 ·

Basic reporting

Multi-view data is data that can be described in many different ways for the same thing, but which all share the same clustering structure. Multi-view clustering problem is an important problem in the community of mechine learning, the authors clearly describe the background and motivation of multi-view clustering. The authors point out the main issues in the current multi-view clustering technique, and propose an efficient technqiue to dear with them. According to the presented Methods section and the experimental results (the source code is provided), the main techniques seem right.

Experimental design

In this paper, the authors consider the multi-view clustering problems. They point out that there are two challenge in the contemporary multi-view clustering community, one is that view-specific representations lack guarantees to reduce noise introduction; and another is that the fusion process compromises view-specific representations, resulting in the inability to capture efficient information from multi-view data. To address these, they propose a unified framework, named AMCFCN, which integrates view-specific encoders, a hybrid attentive module, a deep fusion network, and deep clustering. Compared with seven existing competitive multi-view clustering methods over four data sets, experimental results demonstrate the effectiveness of their approach.

Validity of the findings

A unified framework, named AMCFCN, which integrates view-specific encoders, a hybrid attentive module, a deep fusion network, and deep clustering, is proposed in this paper.The structure of this paper is well organized, and the paper is well written.

Additional comments

However, there are some minor issues that need to be addressed.
1. The case format of each section title should be consistent. For instance, in the paper, “Introduction”, “RELATED WORK”, “A. Formulation of the problem”, “B. hybrid attentive module”, and so on.
2. In the paper, there are many mathematical formulas, some of which are not explained in detail, and the format of the edited mathematical formulas seems “ugly”.
3. There are many places lack of “blank”, which bring symbols or numbers together with words. For example, in lines 436-437,
(I)Deep Canonical Correlation Analysis (DCCA) (Gao et al., 2020);
(II)Deep Canonically Correlated Auto-encoders(DCCAE) (Wang et al., 2015) ;

Cite this review as

Reviewer 2 ·

Basic reporting

In this manuscript, the authors proposed a multi-view clustering network, which integrated attention and contrastive learning strategies and achieved good clustering performances on four public datasets. The ideal is relatively novel and the experiments are well designed. However, the mathematical symbols should be used more normative and the English writing could be improved. In general, the work is interesting and with some novelty. However, the organization of manuscript should be improved. I suggest major revision is need before published.

Experimental design

The work suits the aim of this Journal. The research question is well defined. The investigation performs a relative high technical standard. The code is publicly available.

Validity of the findings

The manuscript proves a novel deep learning-based network for multi-view clustering, which was evaluated on four public dataset. However, there are some unclear descriptions for methods and experiments.

Additional comments

Some improvement could be used for method introduction.
1. The fusion model f(.) and its output G should be put in Fig 1.
2. The detailed structure of view encoder should be given.
3. The mathematical symbols should be used more normative. For example, Line174(L174 for short) and L331, the symbol “V” is defined as “views”, which should not be used as the number of views.
The x in L199 should be X same as defined in (1).
In (2) the equation should be HA(i)=A(Xi), p(xi) is unnecessary here.
L198, T is not defined.
L228, q is same as X in (1)?
L326, P(.) has been used in (6). Change for another symbol.
L326, di,i should be di,j
L325, the range of i and j should be given.
L402, P and T have been used before.
4. The detailed structures of three attention modules should be given in Fig 2.
5. L374, it is unclear that how three views were generated, more details should be given.
6. L403, “map” should be defined before use.
7. L436-L443, the methods compared could be labeled as ref. in Table 3.
8. Fig.7, there are two Coil-20 without Coil -100.
9. There are more typos in the manuscript, please recheck the manuscript carefully.
10. The English writing could be improved.

Cite this review as

---

## Round 0.2 · accepted · Accept

Congrats to the authors for their excellent work. The current version satisfied the reviewers. It can be accepted now.

Reviewer 1 ·

Basic reporting

The authors have modified this paper through comprehensively considering the suggestions. I have no other comments.

Experimental design

The experimental design is complete, and the analysis of experimental results is adequate.

Validity of the findings

The findings in this paper are valuable, and the proposal is novel.

Cite this review as

Reviewer 2 ·

Basic reporting

In this manuscript, the authors proposed a multi-view clustering network, which integrated attention and contrastive learning strategies and achieved good clustering performances on four public datasets. The ideal is relatively novel and the experiments are well designed. The revised version has improved the English writting, Figure, Table and mathematical symbols. All the concerns I have made have been successfully stressed. I suggest this manuscript to be accept by PeerJ computer science.

Experimental design

The work suits the aim of this Journal. The research question is well defined. The investigation performs a relative high technical standard. The code is publicly available.

Validity of the findings

The manuscript proves a novel deep learning-based network for multi-view clustering, which was evaluated on four public data set.The unclear descriptions in last version have been clearly explained.

Additional comments

The revised version has improved the English writting, Figure, Table and mathematical symbols. All the concerns I have made have been successfully stressed. I suggest this manuscript to be accept by PeerJ computer science.

Cite this review as